# Adaptive Maximization of Pointwise Submodular Functions With Budget Constraint

**Nguyen Viet Cuong**[1]     **Huan Xu**[2]

[1]Department of Engineering, University of Cambridge, *vcn22@cam.ac.uk*
[2]Stewart School of Industrial & Systems Engineering, Georgia Institute of Technology,
*huan.xu@isye.gatech.edu*

## Abstract

We study the worst-case adaptive optimization problem with budget constraint that is useful for modeling various practical applications in artificial intelligence and machine learning. We investigate the near-optimality of greedy algorithms for this problem with both modular and non-modular cost functions. In both cases, we prove that two simple greedy algorithms are not near-optimal but the best between them is near-optimal if the utility function satisfies pointwise submodularity and pointwise cost-sensitive submodularity respectively. This implies a combined algorithm that is near-optimal with respect to the optimal algorithm that uses half of the budget. We discuss applications of our theoretical results and also report experiments comparing the greedy algorithms on the active learning problem.

## 1  Introduction

Consider problems where we need to adaptively make a sequence of decisions while taking into account the outcomes of previous decisions. For instance, in the sensor placement problem [1, 2], one needs to sequentially place sensors at some pre-specified locations, taking into account the working conditions of previously deployed sensors. The aim is to cover as large an area as possible while keeping the cost of placement within a given budget. As another example, in the pool-based active learning problem [3, 4], one needs to sequentially select unlabeled examples and query their labels, taking into account the previously observed labels. The aim is to learn a good classifier while ensuring that the cost of querying does not exceed some given budget.

These problems can usually be considered under the framework of *adaptive optimization with budget constraint*. In this framework, the objective is to find a policy for making decisions that maximizes the value of some utility function. With a budget constraint, such a policy must have a cost no higher than the budget given by the problem. Adaptive optimization with budget constraint has been previously studied in the average case [2, 5, 6] and worst case [7]. In this paper, we focus on this problem in the worst case.

In contrast to previous works on adaptive optimization with budget constraint (both in the average and worst cases) [2, 8], we consider not only modular cost functions but also general, possibly non-modular, cost functions on sets of decisions. For example, in the sensor placement problem, the cost of a set of deployed sensors may be the weight of the minimum spanning tree connecting those sensors, where the weight of the edge between any two sensors is the distance between them.[1] In this case, the cost of deploying a sensor is not fixed, but depends on the set of previously deployed sensors. This setting allows the cost function to be non-modular, and thus is more general than the setting in previous works, which usually assume the cost to be modular.

When cost functions are modular, we focus on the useful class of *pointwise submodular* utility functions [2, 7, 8] that has been applied to interactive submodular set cover and active learning problems [7, 8]. With this class of utilities, we investigate the *near-optimality* of greedy policies for worst-case adaptive optimization with budget constraint. A policy is near-optimal if its worst-case utility is within a constant factor of the optimal worst-case utility. We first consider two greedy policies: one that maximizes the worst-case utility gain and one that maximizes the worst-case utility gain per unit cost increment at each step. If the cost is uniform and modular, it is known that these two policies are equivalent and near-optimal [8]; however, we show in this paper that they cannot achieve near-optimality with non-uniform modular costs. Despite this negative result, we can prove that the best between these two greedy policies always achieves near-optimality. This suggests we can combine the two policies into one greedy policy that is near-optimal with respect to the optimal worst-case policy that uses half of the budget. We discuss applications of our theoretical results to the budgeted adaptive coverage problem and the budgeted pool-based active learning problem, both of which can be modeled as worst-case adaptive optimization problems with budget constraint. We also report experimental results comparing the greedy policies on the latter problem.

When cost functions are general and possibly non-modular, we propose a novel class of utility functions satisfying a property called *pointwise cost-sensitive submodularity*. This property is a generalization of *cost-sensitive submodularity* to the adaptive setting. In essence, cost-sensitive submodularity means the utility is more submodular than the cost. Submodularity [9] and pointwise submodularity are special cases of cost-sensitive submodularity and pointwise cost-sensitive submodularity respectively when the cost is modular. With this new class of utilities, we prove similar near-optimality results for the greedy policies as in the case of modular costs. Our proofs build upon the proof techniques for worst-case adaptive optimization with uniform modular costs [8] and non-adaptive optimization with non-uniform modular costs [10] but go beyond them to handle general, possibly non-uniform and non-modular, costs.

## 2 Worst-case Adaptive Optimization with Budget Constraint

We now formalize the framework for worst-case adaptive optimization with budget constraint. Let $\mathcal{X}$ be a finite set of items (or decisions) and $\mathcal{Y}$ be a finite set of possible states (or outcomes). Each item in $\mathcal{X}$ can be in any particular state in $\mathcal{Y}$. Let $h : \mathcal{X} \to \mathcal{Y}$ be a deterministic function that maps each item $x \in \mathcal{X}$ to its state $h(x) \in \mathcal{Y}$. We call $h$ a *realization*. Let $\mathcal{H} \triangleq \mathcal{Y}^{\mathcal{X}} = \{h \mid h : \mathcal{X} \to \mathcal{Y}\}$ be the realization set consisting of all possible realizations.

We consider the problem where we sequentially select a subset of items from $\mathcal{X}$ as follows: we select an item, observe its state, then select the next item, observe its state, etc. After some iterations, our observations so far can be represented as a *partial realization*, which is a partial function from $\mathcal{X}$ to $\mathcal{Y}$. An *adaptive* strategy to select items takes into account the states of all previous items when deciding the next item to select. Each adaptive strategy can be encoded as a deterministic policy for selecting items, where a policy is a function from a partial realization to the next item to select. A policy can be represented by a policy tree in which each node is an item to be selected and edges below a node correspond to its states.

We assume there is a cost function $c : 2^{\mathcal{X}} \to \mathbb{R}_{\geq 0}$, where $2^{\mathcal{X}}$ is the power set of $\mathcal{X}$. For any set of items $S \subseteq \mathcal{X}$, $c(S)$ is the cost incurred if we select the items in $S$ and observe their states. For simplicity, we also assume $c(\emptyset) = 0$ and $c(S) > 0$ for $S \neq \emptyset$. If $c$ is modular, then $c(S) = \sum_{x \in S} c(\{x\})$ for all $S$. In general, $c$ can be non-modular. We shall consider the modular cost setting in Section 3 and the non-modular cost setting in Section 4.

For a policy $\pi$, we define the cost of $\pi$ as the maximum cost incurred by a set of items selected along any path of the policy tree of $\pi$. Note that if we fix a realization $h$, the set of items selected by the policy $\pi$ is fixed, and we denote this set by $x_h^{\pi}$. The set $x_h^{\pi}$ corresponds to a path of the policy tree of $\pi$, and thus the cost of $\pi$ can be formally defined as $c(\pi) \triangleq \max_{h \in \mathcal{H}} c(x_h^{\pi})$.

In the worst-case adaptive optimization problem, we have a utility function $f : 2^{\mathcal{X}} \times \mathcal{H} \to \mathbb{R}_{\geq 0}$ that we wish to maximize in the worst case. The utility function $f(S, h)$ depends on a set $S$ of selected items and a realization $h$ that determines the states of all items. Essentially, $f(S, h)$ denotes the value of selecting $S$, given that the true realization is $h$. We assume $f(\emptyset, h) = 0$ for all $h$.

For a policy $\pi$, we define its worst-case utility as $f_{\text{worst}}(\pi) \triangleq \min_{h \in \mathcal{H}} f(x_h^{\pi}, h)$. Given a budget $K > 0$, our goal is to find a policy $\pi^*$ whose cost does not exceed $K$ and $\pi^*$ maximizes $f_{\text{worst}}$.

Formally, $\pi^* \triangleq \arg\max_\pi f_{\text{worst}}(\pi)$ subject to $c(\pi) \leq K$. We call this the problem of *worst-case adaptive optimization with budget constraint*.

# 3 Modular Cost Setting

In this section, we consider the setting where the cost function is modular. This setting is very common in the literature (e.g., see [2, 10, 11, 12]). We will describe the assumptions on the utility function, the greedy algorithms for worst-case adaptive optimization with budget constraint, and the analyses of these algorithms. Proofs in this section are given in the supplementary material.

## 3.1 Assumptions on the Utility Function

Adaptive optimization with an arbitrary utility function is often infeasible, so we only focus on a useful class of utility functions: the pointwise monotone submodular functions. Recall that a set function $g : 2^{\mathcal{X}} \to \mathbb{R}$ is submodular if it satisfies the following diminishing return property: for all $A \subseteq B \subseteq \mathcal{X}$ and $x \in \mathcal{X} \setminus B$, $g(A \cup \{x\}) - g(A) \geq g(B \cup \{x\}) - g(B)$. Furthermore, $g$ is monotone if $g(A) \leq g(B)$ for all $A \subseteq B$. In our setting, the utility function $f(S, h)$ depends on both the selected items and the realization, and we assume it satisfies the *pointwise submodularity*, *pointwise monotonicity*, and *minimal dependency* properties below.

**Definition 1** (Pointwise Submodularity). *A utility function $f(S, h)$ is pointwise submodular if the set function $f_h(S) \triangleq f(S, h)$ is submodular for all $h \in \mathcal{H}$.*

**Definition 2** (Pointwise Monotonicity). *A utility function $f(S, h)$ is pointwise monotone if the set function $f_h(S) \triangleq f(S, h)$ is monotone for all $h \in \mathcal{H}$.*

**Definition 3** (Minimal Dependency). *A utility function $f(S, h)$ satisfies minimal dependency if the value of $f(S, h)$ only depends on the items in $S$ and their states (with respect to the realization $h$).*

These properties are useful for worst-case adaptive optimization and were also considered in [8] for uniform modular costs. Pointwise submodularity is an extension of submodularity and pointwise monotonicity is an extension of monotonicity to the adaptive setting. Minimal dependency is needed to make sure the value of $f$ only depends on what have already been observed. Without this property, the value of $f$ may be unpredictable and is hard to be reasoned about. The three assumptions above hold for practical utility functions that we will describe in Section 5.1.

## 3.2 Greedy Algorithms and Theoretical Results

Our paper focuses on greedy algorithms (or greedy policies) to maximize the worst-case utility with a budget constraint. We are interested in a theoretical guarantee for these policies: the *near-optimality* guarantee. Specifically, a policy is near-optimal if its worst-case utility is within a constant factor of the optimal worst-case utility. In this section, we consider two intuitive greedy policies and prove that each of these policies is individually not near-optimal but the best between them will always be near-optimal. We shall also discuss a combined policy and its guarantee in this section.

### 3.2.1 Two Greedy Policies

We consider two greedy policies in Figure 1. These policies are described in the general form and can be used for both modular and non-modular cost functions. In these policies, $\mathcal{D}$ is the partial realization that we have observed so far, and $X_{\mathcal{D}} \triangleq \{x \in \mathcal{X} \mid (x, y) \in \mathcal{D} \text{ for some } y \in \mathcal{Y}\}$ is the domain of $\mathcal{D}$ (i.e., the set of selected items in $\mathcal{D}$). For any item $x$, we write $\delta(x \mid \mathcal{D})$ to denote the worst-case utility gain if $x$ is selected after we observe $\mathcal{D}$. That is,

$$\delta(x \mid \mathcal{D}) \triangleq \min_{y \in \mathcal{Y}} \{f(X_{\mathcal{D}} \cup \{x\}, \mathcal{D} \cup \{(x, y)\}) - f(X_{\mathcal{D}}, \mathcal{D})\}. \tag{1}$$

In this definition, note that we have extended the utility function $f$ to take a partial realization as the second parameter (instead of a full realization). This extension is possible because the utility function is assumed to satisfy minimal dependency, and thus its value only depends on the partial realization that we have observed so far. In the policy $\pi_1$, for any item $x \in \mathcal{X}$ and any $S \subseteq \mathcal{X}$, we define:

$$\Delta c(x \mid S) \triangleq c(S \cup \{x\}) - c(S), \tag{2}$$

which is the cost increment of selecting $x$ after $S$ has been selected. If the cost function $c$ is modular, then $\Delta c(x \mid S) = c(\{x\})$.

| Cost-average Greedy Policy $\pi_1$: | Cost-insensitive Greedy Policy $\pi_2$: |
|---|---|
| $\mathcal{D} \leftarrow \emptyset; \quad U \leftarrow \mathcal{X};$ | $\mathcal{D} \leftarrow \emptyset; \quad U \leftarrow \mathcal{X};$ |
| **repeat** | **repeat** |
| $\quad$ Pick $x^* \in U$ that maximizes $\delta(x^* \mid \mathcal{D})/\Delta c(x^* \mid X_{\mathcal{D}})$; | $\quad$ Pick $x^* \in U$ that maximizes $\delta(x^* \mid \mathcal{D})$; |
| $\quad$ **if** $c(X_{\mathcal{D}} \cup \{x^*\}) \leq K$ **then** | $\quad$ **if** $c(X_{\mathcal{D}} \cup \{x^*\}) \leq K$ **then** |
| $\quad\quad$ Observe state $y^*$ of $x^*$; | $\quad\quad$ Observe state $y^*$ of $x^*$; |
| $\quad\quad$ $\mathcal{D} \leftarrow \mathcal{D} \cup \{(x^*, y^*)\}$; | $\quad\quad$ $\mathcal{D} \leftarrow \mathcal{D} \cup \{(x^*, y^*)\}$; |
| $\quad$ **end** | $\quad$ **end** |
| $\quad$ $U \leftarrow U \setminus \{x^*\}$; | $\quad$ $U \leftarrow U \setminus \{x^*\}$; |
| **until** $U = \emptyset$; | **until** $U = \emptyset$; |

Figure 1: Two greedy policies for adaptive optimization with budget constraint.

The two greedy policies in Figure 1 are intuitive. The cost-average policy $\pi_1$ greedily selects the items that maximize the worst-case utility gain per unit cost increment if they are still affordable by the remaining budget. On the other hand, the cost-insensitive policy $\pi_2$ simply ignores the items' costs and greedily selects the affordable items that maximize the worst-case utility gain.

**Analyses of $\pi_1$ and $\pi_2$:** Given the two greedy policies, we are interested in their near-optimality: whether they provide a constant factor approximation to the optimal worst-case utility. Unfortunately, we can show that these policies are not near-optimal. This negative result is stated in Theorem 1 below. The proof of this theorem constructs counter-examples where the policies are not near-optimal.

**Theorem 1.** *For any $\pi_i \in \{\pi_1, \pi_2\}$ and $\alpha > 0$, there exists a worst-case adaptive optimization problem with a utility $f$, a modular cost $c$, and a budget $K$ such that $f$ satisfies the assumptions in Section 3.1 and $f_{worst}(\pi_i)/f_{worst}(\pi^*) < \alpha$, where $\pi^*$ is the optimal policy for the problem.*

### 3.2.2 A Near-optimal Policy

Although the greedy policies $\pi_1$ and $\pi_2$ are not near-optimal, we now show that the best between them is in fact near-optimal. More specifically, let us define a policy $\pi$ such that:

$$\pi \triangleq \begin{cases} \pi_1 & \text{if } f_{\text{worst}}(\pi_1) > f_{\text{worst}}(\pi_2) \\ \pi_2 & \text{otherwise} \end{cases}. \tag{3}$$

Theorem 2 below states that $\pi$ is near-optimal for the worst-case adaptive optimization problem with budget constraint.

**Theorem 2.** *Let $f$ be a utility that satisfies the assumptions in Section 3.1 and $\pi^*$ be the optimal policy for the worst-case adaptive optimization problem with utility $f$, a modular cost $c$, and a budget $K$. The policy $\pi$ defined by Equation (3) satisfies $f_{worst}(\pi) > \frac{1}{2}(1 - 1/e) f_{worst}(\pi^*)$.*

The constant factor $\frac{1}{2}(1 - 1/e)$ in Theorem 2 is slightly worse than the constant factor $(1 - 1/\sqrt{e})$ for the non-adaptive budgeted maximum coverage problem [10]. If we apply this theorem to a problem with a uniform cost, i.e., $c(\{x\}) = c(\{x'\})$ for all $x$ and $x'$, then $\pi_1 = \pi_2$ and $f_{\text{worst}}(\pi) = f_{\text{worst}}(\pi_1) = f_{\text{worst}}(\pi_2)$. Thus, from Theorem 2, $f_{\text{worst}}(\pi_1) = f_{\text{worst}}(\pi_2) > \frac{1}{2}(1 - 1/e) f_{\text{worst}}(\pi^*)$. Although this implies the greedy policy is near-optimal, the constant factor $\frac{1}{2}(1 - 1/e)$ in this case is not as good as the constant factor $(1 - 1/e)$ in [8] for the uniform modular cost setting. We also note that Theorem 2 still holds if we replace the cost-insensitive policy $\pi_2$ with only the first item that it selects (see its proof for details). In other words, we can terminate $\pi_2$ right after it selects the first item and the near-optimality in Theorem 2 is still guaranteed.

### 3.2.3 A Combined Policy

With Theorem 2, a naive approach to the worst-case adaptive optimization problem with budget constraint is to estimate $f_{\text{worst}}(\pi_1)$ and $f_{\text{worst}}(\pi_2)$ (without actually running these policies) and use the best between them. However, exact estimation of these quantities is intractable because it would require a consideration of all realizations (an exponential number of them) to find the worst-case realization for these policies. This is very different from the non-adaptive setting [10, 12, 13] where we can easily find the best policy because there is only one realization.

Furthermore, in the adaptive setting, we cannot roll back once we run a policy. For example, we cannot run $\pi_1$ and $\pi_2$ at the same time to determine which one is better without doubling the budget.

This is because we have to pay the cost every time we want to observe the state of an item, and the next item selected would depend on the previous states. Thus, the adaptive setting in our paper is more difficult than the non-adaptive setting considered in previous works [10, 12, 13]. If we consider a Bayesian setting with some prior on the set of realizations [2, 4, 14], we can sample a subset of realizations from the prior to estimate $f_{worst}$. However, this method does not provide any guarantee for the estimation.

Given these difficulties, a more practical approach is to run both $\pi_1$ and $\pi_2$ using half of the budget for each policy and combine the selected sets. Details of this combined policy ($\pi_{1/2}$) are in Figure 2. Using Theorem 2, we can show that $\pi_{1/2}$ is near-optimal compared to the optimal worst-case policy that uses half of the budget. Theorem 3 below states this result. We note that the theorem still holds if the order of running $\pi_1$ and $\pi_2$ is exchanged in the policy $\pi_{1/2}$.

1. Run $\pi_1$ with budget $K/2$ (half of the total budget), and let the set of selected items be $S_1$.
2. Starting with the empty set, run $\pi_2$ with budget $K/2$ and let the set of items selected in this step be $S_2$. For simplicity, we allow $S_2$ to overlap with $S_1$.
3. Return $S_1 \cup S_2$.

Figure 2: The combined policy $\pi_{1/2}$.

**Theorem 3.** *Assume the same setting as in Theorem 2. Let $\pi_{1/2}^*$ be the optimal policy for the worst-case adaptive optimization problem with budget $K/2$. The policy $\pi_{1/2}$ satisfies $f_{worst}(\pi_{1/2}) > \frac{1}{2}\left(1 - 1/e\right) f_{worst}(\pi_{1/2}^*).$*

Since Theorem 3 only compares $\pi_{1/2}$ with the optimal policy $\pi_{1/2}^*$ that uses half of the budget, a natural question is whether or not the policies $\pi_1$ and $\pi_2$ running with the full budget have a similar guarantee compared to $\pi_{1/2}^*$. Using the same counter-example for $\pi_2$ in the proof of Theorem 1, we can easily show in Theorem 4 that this guarantee does not hold for the cost-insensitive policy $\pi_2$.

**Theorem 4.** *For any $\alpha > 0$, there exists a worst-case adaptive optimization problem with a utility $f$, a modular cost $c$, and a budget $K$ such that $f$ satisfies the assumptions in Section 3.1 and $f_{worst}(\pi_2)/f_{worst}(\pi_{1/2}^*) < \alpha$, where $\pi_{1/2}^*$ is the optimal policy for the problem with budget $K/2$.*

As regards the cost-average policy $\pi_1$, it remains open whether running it with full budget provides any constant factor approximation to the worst-case utility of $\pi_{1/2}^*$. However, in the supplementary material, we show that it is not possible to construct a counter-example for this case using a modular utility function, so a counter-example (if there is any) should use a more sophisticated utility.

## 4 Non-Modular Cost Setting

We first define cost-sensitive submodularity, a generalization of submodularity that takes into account a general, possibly non-modular, cost on sets of items. We then state the assumptions on the utility function and the near-optimality results of the greedy algorithms for this setting.

**Cost-sensitive Submodularity:** Let $c$ be a general cost function that is strictly monotone, i.e., $c(A) < c(B)$ for all $A \subset B$. Hence, $\Delta c(x \mid S) > 0$ for all $S$ and $x \notin S$. Assume $c$ satisfies the triangle inequality: $c(A \cup B) \leq c(A) + c(B)$ for all $A, B \subseteq \mathcal{X}$. We define cost-sensitive submodularity as follows.

**Definition 4** (Cost-sensitive Submodularity). *A set function $g : 2^{\mathcal{X}} \to \mathbb{R}$ is cost-sensitively submodular w.r.t. a cost function $c$ if it satisfies: for all $A \subseteq B \subseteq \mathcal{X}$ and $x \in \mathcal{X} \setminus B$,*

$$\frac{g(A \cup \{x\}) - g(A)}{\Delta c(x \mid A)} \geq \frac{g(B \cup \{x\}) - g(B)}{\Delta c(x \mid B)}. \tag{4}$$

In essence, cost-sensitive submodularity is a generalization of submodularity and means that $g$ is more submodular than the cost $c$. When $c$ is modular, cost-sensitive submodularity is equivalent to submodularity. If $g$ is cost-sensitively submodular w.r.t. a submodular cost, it will also be submodular. Since $c$ satisfies the triangle inequality, it cannot be super-modular but it can be non-submodular (see the supplementary for an example).

We state some useful properties of cost-sensitive submodularity in Theorem 5. In this theorem, $\alpha g_1 + \beta g_2$ is the function $g(S) = \alpha g_1(S) + \beta g_2(S)$ for all $S \subseteq \mathcal{X}$, and $\alpha c_1 + \beta c_2$ is the function $c(S) = \alpha c_1(S) + \beta c_2(S)$ for all $S \subseteq \mathcal{X}$. The proof of this theorem is in the supplementary material.

**Theorem 5.** *(a) If $g_1$ and $g_2$ are cost-sensitively submodular w.r.t. a cost function c, then $\alpha g_1 + \beta g_2$ is also cost-sensitively submodular w.r.t. c for all $\alpha, \beta \geq 0$.*
*(b) If g is cost-sensitively submodular w.r.t. cost functions $c_1$ and $c_2$, then g is also cost-sensitively submodular w.r.t. $\alpha c_1 + \beta c_2$ for all $\alpha, \beta \geq 0$ such that $\alpha + \beta > 0$.*
*(c) For any integer $n \geq 1$, if g is monotone and $c(S) = \sum_{i=1}^{n} a_i (g(S))^i$ with non-negative coefficients $a_i \geq 0$ such that $\sum_{i=1}^{n} a_i > 0$, then g is cost-sensitively submodular w.r.t. c.*
*(d) If g is monotone and $c(S) = \alpha e^{g(S)}$ for $\alpha > 0$, then g is cost-sensitively submodular w.r.t. c.*

This theorem specifies various cases where a function $g$ is cost-sensitively submodular w.r.t. a cost $c$. Note that neither $g$ nor $c$ needs to be submodular for this theorem to hold. Parts (a,b) state that cost-sensitive submodularity is preserved for linear combinations of either $g$ or $c$. Parts (c,d) state that if $c$ is a polynomial (respectively, exponential) of $g$ with non-negative (respectively, positive) coefficients, then $g$ is cost-sensitively submodular w.r.t. $c$.

**Assumptions on the Utility:** In this setting, we also assume the utility $f(S, h)$ satisfies pointwise monotonicity and minimal dependency. Furthermore, we assume it satisfies the pointwise cost-sensitive submodularity property below. This property is an extension of cost-sensitive submodularity to the adaptive setting and is also a generalization of pointwise submodularity for a general cost. If the cost is modular, pointwise cost-sensitive submodularity is equivalent to pointwise submodularity.

**Definition 5** (Pointwise Cost-sensitive Submodularity). *A utility $f(S, h)$ is pointwise cost-sensitively submodular w.r.t. a cost c if, for all h, $f_h(S) \triangleq f(S, h)$ is cost-sensitively submodular w.r.t. c.*

**Theoretical Results:** Under the above assumptions, near-optimality guarantees in Theorems 2 and 3 for the greedy algorithms in Section 3.2 still hold. This result is stated and proven in the supplementary material. The proof requires a sophisticated combination of the techniques for worst-case adaptive optimization with uniform modular costs [8] and non-adaptive optimization with non-uniform modular costs [10]. Unlike [10], our proof deals with policy trees instead of sets and we generalize previous techniques, originally used for modular costs, to handle general cost functions.

# 5 Applications and Experiments

## 5.1 Applications

We discuss two applications of our theoretical results in this section: the budgeted adaptive coverage problem and the budgeted pool-based active learning problem. These problems were considered in [2] for the average case, while we study them here in the worst case where the difficulty, as shown above, is that simple policies such as $\pi_1$ and $\pi_2$ are not near-optimal as compared to the former case.

**Budgeted Adaptive Coverage:** In this problem, we are given a set of locations where we need to place some sensors to get the spatial information of the surrounding environment. If sensors are deployed at a set of sensing locations, we have to pay a cost depending on where the locations are. After a sensor is deployed at a location, it may be in one of a few possible states (e.g., this may be caused by a partial failure of the sensor), leading to various degrees of information covered by the sensor. The *budgeted adaptive coverage* problem can be stated as: given a cost budget $K$, where should we place the sensors to cover as much spatial information as possible?

We can model this problem as a worst-case adaptive optimization problem with budget $K$. Let $\mathcal{X}$ be the set of all possible locations where sensors may be deployed, and let $\mathcal{Y}$ be the set of all possible states of the sensors. For each set of locations $S \subseteq \mathcal{X}$, $c(S)$ is the cost of deploying sensors there. For a location $x$ and a state $y$, let $R_{x,y}$ be the geometric shape associated with the spatial information covered if we put a sensor at $x$ and its state is $y$. We can define the utility function $f(S, h) = |\bigcup_{x \in S} R_{x,h(x)}|$, which is the cardinality (or volume) of the covered region. If we fix a realization $h$, this utility is monotone submodular [11]. Thus, $f(S, h)$ is pointwise monotone submodular. Since this function also satisfies minimal dependency, we can apply the policy $\pi_{1/2}$ to this problem and get the guarantee in Theorem 3 if the cost function $c$ is modular.

**Budgeted Pool-based Active Learning:** For pool-based active learning, we are given a finite set of unlabeled examples and need to adaptively query the labels of some selected examples from that set to train a classifier. Every time we query an example, we have to pay a cost and then get to see its label. In the next iteration, we can use the labels observed so far to select the next example to

Table 1: AUCs (normalized to [0,100]) of four learning policies.

| Cost | Data set 1 | | | | Data set 2 | | | | Data set 3 | | | |
|---|---|---|---|---|---|---|---|---|---|---|---|---|
| | PL | LC | ALC | BLC | PL | LC | ALC | BLC | PL | LC | ALC | BLC |
| R1 | 79.8 | 85.6 | **93.9** | 92.0 | 69.0 | 69.3 | **83.1** | 77.5 | 76.7 | 79.7 | **94.0** | 90.1 |
| R2 | 80.7 | **85.0** | 63.0 | 63.6 | **70.9** | 70.4 | 50.5 | 51.8 | 78.6 | **82.6** | 51.9 | 54.7 |
| M1 | 92.5 | 93.0 | **96.5** | 95.9 | 84.6 | 86.7 | 91.7 | **92.6** | 90.7 | 91.0 | **96.9** | 96.3 |
| M2 | 86.9 | 87.4 | **91.2** | 90.1 | 72.5 | **73.1** | 62.1 | 67.4 | 79.4 | **86.3** | 74.1 | 78.2 |

query. The *budgeted pool-based active learning* problem can be stated as: given a budget $K$, which examples should we query to train a good classifier?

We can model this problem as a worst-case adaptive optimization problem with budget $K$. Let $\mathcal{X}$ be the set of unlabeled examples and $\mathcal{Y}$ be the set of all possible labels. For each set of examples $S \subseteq \mathcal{X}$, $c(S)$ is the cost of querying their labels. A realization $h$ is a labeling of all examples in $\mathcal{X}$. For pool-based active learning, previous works [2, 8, 14] have shown that the version space reduction utility is pointwise monotone submodular and satisfies minimal dependency. This utility is defined as $f(S, h) = \sum_{h':h'(S) \neq h(S)} p_0[h']$, where $p_0$ is a prior on $\mathcal{H}$ and $h(S)$ is the labels of $S$ according to $h$. Thus, we can apply $\pi_{1/2}$ to this problem with the guarantee in Theorem 3 if the cost $c$ is modular.

With the utility above, the greedy criterion that maximizes $\delta(x^* \mid \mathcal{D})$ in the cost-insensitive policy $\pi_2$ is equivalent to the well-known least confidence criterion $x^* = \arg\min_x \max_y p_{\mathcal{D}}[y; x] = \arg\max_x \min_y \{1 - p_{\mathcal{D}}[y; x]\}$, where $p_{\mathcal{D}}$ is the posterior after observing $\mathcal{D}$ and $p_{\mathcal{D}}[y; x]$ is the probability that $x$ has label $y$. On the other hand, the greedy criterion that maximizes $\delta(x^* \mid \mathcal{D})/\Delta c(x^* \mid X_{\mathcal{D}})$ in the cost-average policy $\pi_1$ is equivalent to:

$$x^* = \arg\max_x \left\{ \frac{\min_y \{1 - p_{\mathcal{D}}[y; x]\}}{\Delta c(x \mid X_{\mathcal{D}})} \right\}. \tag{5}$$

We prove this equation in the supplementary material. Theorem 3 can also be applied if we consider the total generalized version space reduction utility [8] that incorporates an arbitrary loss. This utility was also shown to be pointwise monotone submodular and satisfy minimal dependency [8], and thus the theorem still holds in this case for modular costs.

## 5.2 Experiments

We present experimental results for budgeted pool-based active learning with various modular cost settings. We use 3 binary classification data sets extracted from the 20 Newsgroups data [15]: alt.atheism/comp.graphics (data set 1), comp.sys.mac.hardware/comp.windows.x (data set 2), and rec.motorcycles/rec.sport.baseball (data set 3). Since the costs are modular, they are put on individual examples, and the total cost is the sum of the selected examples' costs. We will consider settings where random costs and margin-dependent costs are put on training data.

We compare 4 data selection strategies: passive learning (PL), cost-insensitive greedy policy or least confidence (LC), cost-average greedy policy (ALC), and budgeted least confidence (BLC). LC and ALC have been discussed in Section 5.1, and BLC is the corresponding policy $\pi_{1/2}$. These three strategies are active learning algorithms. For comparison, we train a logistic regression model with budgets 50, 100, 150, and 200, and approximate its area under the learning curve (AUC) using the accuracies on a separate test set. In Table 1, bold numbers indicate the best scores, and underlines indicate that BLC is the second best among the active learning algorithms.

**Experiments with Random Costs:** In this setting, costs are put randomly to the training examples in 2 scenarios. In scenario R1, some random examples have a cost drawn from Gamma(80, 0.1) and the other examples have cost 1. From the results for this scenario in Table 1, ALC is better than LC and BLC is the second best among the active learning algorithms. In scenario R2, all examples with label 1 have a cost drawn from Gamma(45, 0.1) and the others (examples with label 0) have cost 1. From Table 1, LC is better than ALC in this scenario, which is due to the biasness of ALC toward examples with label 0. In this scenario, BLC is also the second best among the active learning algorithms, although it is still significantly worse than LC.

**Experiments with Margin-Dependent Costs:** In this setting, costs are put on training examples based on their margins to a classifier trained on the whole data set. Specifically, we first train a logistic regression model on all the data and compute its probabilistic prediction for each training example.

The margin of an example is then the scaled distance between 0.5 and its probabilistic prediction. We also consider 2 scenarios. In scenario M1, we put higher costs on examples with lower margins. From Table 1, ALC is better than LC in this scenario. BLC performs better than both ALC and LC on data set 2, and performs the second best among the active learning algorithms on data sets 1 and 3. In scenario M2, we put higher costs on examples with larger margins. From Table 1, ALC is better than LC on data set 1, while LC is better than ALC on data sets 2 and 3. On all data sets, BLC is the second best among the active learning algorithms.

Note that our experiments do not intend to show BLC is better than LC and ALC. In fact, our theoretical results somewhat state that either LC or ALC will perform well although we may not know which one is better. So, our experiments are to demonstrate some cases where one of these methods would perform badly, and BLC can be a more robust choice that often performs in-between these two methods.

## 6 Related Work

Our work is related to [7, 8, 10, 12] but is more general than these works. Cuong et al. [8] considered a similar worst-case setting as ours, but they assumed the utility is pointwise submodular and the cost is uniform modular. Our work is more general than theirs in two aspects: (1) pointwise cost-sensitive submodularity is a generalization of pointwise submodularity, and (2) our cost function is general and may be neither uniform nor modular. These generalizations make the problem more complicated as simple greedy policies, which are near-optimal in [8], will not be near-optimal anymore (see Section 3.2). Thus, we need to combine two simple greedy policies to obtain a new near-optimal policy.

Guillory & Bilmes [7] were the first to consider worst-case adaptive submodular optimization, particularly in the interactive submodular set cover problem [7, 16]. In [7], the utility is also pointwise submodular, and they look for a policy that can achieve at least a certain value of utility w.r.t. an unknown target realization while at the same time minimizing the cost of this policy. Their final utility, which is derived from the individual utilities of various realizations, is submodular. Our work, in contrast, tries to maximize the worst-case utility directly given a cost budget.

Khuller et al. [10] considered the budgeted maximum coverage problem, which is the non-adaptive version of our problem with a modular cost. For this problem, they showed that the best between two non-adaptive greedy policies can achieve near-optimality compared to the optimal non-adaptive policy. Similar results were also shown in [13] with a better constant and in [12] for the outbreak detection problem. Our work is a generalization of [10, 12] to the adaptive setting with general cost functions, and we can achieve the same constant factor as [12]. Furthermore, the class of utility functions in our work is even more general than the coverage utilities in these works.

Our concept of cost-sensitive submodularity is a generalization of submodularity [9] for general costs. Submodularity has been successfully applied to many applications [1, 17, 18, 19, 20]. Besides pointwise submodularity, there are other ways to extend submodularity to the adaptive setting, e.g., adaptive submodularity [2, 21, 22] and approximately adaptive submodularity [23]. For adaptive submodular utilities, Golovin & Krause [2] proved that greedily maximizing the average utility gain in each step is near-optimal in both average and worst cases. However, neither pointwise submodularity implies adaptive submodularity nor vice versa. Thus, our assumptions in this paper can be applied to a different class of utilities than those in [2].

## 7 Conclusion

We studied worst-case adaptive optimization with budget constraint, where the cost can be either modular or non-modular and the utility satisfies pointwise submodularity or pointwise cost-sensitive submodularity respectively. We proved a negative result about two greedy policies for this problem but also showed a positive result for the best between them. We used this result to derive a combined policy which is near-optimal compared to the optimal policy that uses half of the budget. We discussed applications of our theoretical results and reported experiments for the greedy policies on the pool-based active learning problem.

**Acknowledgments**

This work was done when both authors were at the National University of Singapore. The authors were partially supported by the Agency for Science, Technology and Research (A*STAR) of Singapore through SERC PSF Grant R266000101305.

## Footnotes

[1]This cost function is reasonable in practice if we think of it as the minimal necessary communication cost to keep the sensors connected (rather than the placement cost).

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
