[Supplementary Material]

# Supplementary Material:
# Adaptive Maximization of Pointwise Submodular Functions With Budget Constraint

**Nguyen Viet Cuong**[1]          **Huan Xu**[2]

[1]Department of Engineering, University of Cambridge, *vcn22@cam.ac.uk*

[2]Stewart School of Industrial & Systems Engineering, Georgia Institute of Technology, *huan.xu@isye.gatech.edu*

## A    Proof of Theorem 1

We prove Theorem 1 for the cost-average greedy policy $\pi_1$ and the cost-insensitive greedy policy $\pi_2$ below. For each policy, we construct a worst-case adaptive optimization problem that satisfies the theorem. In this problem, the utility and cost are both modular, i.e., they can be decomposed into the sum of the utilities (or costs) of the individual items. Besides, all the items have only one state, so it is essentially a non-adaptive problem.

### A.1    Cost-average Greedy Policy $\pi_1$

Consider the utility function:

$$f(S, h) = \sum_{x \in S} w(x, h(x)), \tag{A.1}$$

where $w : \mathcal{X} \times \mathcal{Y} \to \mathbb{R}_{\geq 0}$ is the utility function for one item. Intuitively, $w(x, y)$ is the utility obtained by selecting item $x$ with state $y$, and $f(S, h)$ is the sum of all the utilities of the items in $S$ with states according to $h$. It is easy to see that $f$ is pointwise submodular, pointwise monotone, and also satisfies minimal dependency.

We also consider the worst-case adaptive optimization problem with two items $\{x_1, x_2\}$ and one state $\{0\}$ such that $w(x_1, 0) = 1$ and $w(x_2, 0) = p$, for some $p > 1$. Let the cost function be:

$$c(\emptyset) = 0, \quad c(\{x_1\}) = 1, \quad c(\{x_2\}) = p + 1, \quad c(\{x_1, x_2\}) = c(\{x_1\}) + c(\{x_2\}) = p + 2,$$

and let the budget be $K = p + 1$. With this budget, a policy is only allowed to select at most one item.

For this problem, the policy $\pi_1$ would select the item $x_1$ because:

$$\frac{\delta(x_1 \mid \emptyset)}{c(\{x_1\})} = \frac{\min_y f(\{x_1\}, \{(x_1, y)\})}{c(\{x_1\})} = 1 > \frac{p}{p + 1} = \frac{\min_y f(\{x_2\}, \{(x_2, y)\})}{c(\{x_2\})} = \frac{\delta(x_2 \mid \emptyset)}{c(\{x_2\})}.$$

Thus, $f_{\text{worst}}(\pi_1) = 1$. However, the optimal policy $\pi^*$ would select $x_2$ to obtain $f_{\text{worst}}(\pi^*) = p$. Hence, $f_{\text{worst}}(\pi_1)/f_{\text{worst}}(\pi^*) = 1/p$. By increasing $p$, we can have $f_{\text{worst}}(\pi_1)/f_{\text{worst}}(\pi^*) < \alpha$ for any $\alpha > 0$.

### A.2    Cost-insensitive Greedy Policy $\pi_2$

Consider the worst-case adaptive optimization problem with $n + 1$ items $\{x_0, x_1, \ldots, x_n\}$ and one state $\{0\}$. We will also use the utility function $f$ defined by Equation (A.1) above with $w(x_0, 0) = 2$ and $w(x_i, 0) = 1$ for $i = 1, \ldots, n$. This utility satisfies the assumptions in Theorem 1. Let the cost function be:

$$c(\{x_0\}) = n, \quad c(\{x_i\}) = 1 \text{ for } i = 1, \ldots, n \quad \text{and } c(S) = \sum_{x \in S} c(\{x\}) \text{ for other subsets of items } S.$$

We let the budget be $K = n$. With this budget, a policy may select exactly one item $x_0$, or it may ignore $x_0$ and select only the items among $\{x_1, \ldots, x_n\}$.

For this problem, the policy $\pi_2$ would select the item $x_0$ because for any $i = 1, \ldots, n$:

$$\delta(x_0 \mid \emptyset) = \min_y f(\{x_0\}, \{(x_0, y)\}) = 2 > 1 = \min_y f(\{x_i\}, \{(x_i, y)\}) = \delta(x_i \mid \emptyset).$$

Thus, $f_{\text{worst}}(\pi_2) = 2$. However, the optimal policy $\pi^*$ would select all the items $\{x_1, \ldots, x_n\}$ to obtain $f_{\text{worst}}(\pi^*) = n$. Hence, $f_{\text{worst}}(\pi_2)/f_{\text{worst}}(\pi^*) = 2/n$. By increasing $n$, we can have $f_{\text{worst}}(\pi_2)/f_{\text{worst}}(\pi^*) < \alpha$ for any $\alpha > 0$.

# B    Proof of Theorems 2, 3, 4, and Discussion on $\pi_1$ versus $\pi^*_{1/2}$

Theorems 2 and 3 are special cases of Theorems F.1 and F.2 in Section F respectively (see that section for the general theorem statements and proofs). The proof of Theorem 4 uses the same counter-example for policy $\pi_2$ in Section A above.

We now give a discussion on $\pi_1$ versus $\pi^*_{1/2}$. In particular, we show that it is not possible to construct a counter-example for the policy $\pi_1$ with full budget compared to $\pi^*_{1/2}$ if we use the simple utility and modular cost functions in the proof of Theorem 1 above. This means we will prove that $\pi_1$ provides a constant factor approximation to $\pi^*_{1/2}$ for those utility and modular cost functions. We state and prove this result in the proposition below.

**Proposition B.1.** *For any utility function $f(S, h) \triangleq \sum_{x \in S} w(x, h(x))$ where $w : \mathcal{X} \times \mathcal{Y} \to \mathbb{R}_{\geq 0}$, and any modular cost function $c$ such that $c(S) = \sum_{x \in S} c(\{x\})$,*

$$f_{\text{worst}}(\pi_1) > \frac{1}{2}\left(1 - \frac{1}{e}\right) f_{\text{worst}}(\pi^*_{1/2}),$$

*where $\pi_1$ is run with budget $K$ and $\pi^*_{1/2}$ is the optimal worst-case policy with budget $K/2$.*

*Proof.* For this utility, note that the realization $h^*(x) \triangleq \arg\min_y w(x, y)$ is always the worst-case realization of *any* policy. Besides, $\delta(x \mid \mathcal{D}) = w(x, h^*(x))$, which means the greedy criterion in policy $\pi_1$ would always consider the state $h^*(x)$ instead of other states. So, we can fix the realization $h^*$ in all of our following arguments.

Assume we run $\pi_1$ with budget $K/2$ and select $x'_1, x'_2, \ldots, x'_t, x'_{t+1}, \ldots, x'_T$, while at the same time we run $\pi_1$ with budget $K$ and select $x'_1, x'_2, \ldots, x'_t, x_{t+1}$, where $x_{t+1}$ is the first item selected by $\pi_1$ with budget $K$ but could not be selected with budget $K/2$ due to the budget constraint. From the greedy criterion of $\pi_1$, it is easy to see that:

$$\frac{w(x_{t+1}, h^*(x_{t+1}))}{c(\{x_{t+1}\})} \geq \frac{w(x'_i, h^*(x'_i))}{c(\{x'_i\})}, \text{ for } i = t+1, \ldots, T.$$

Thus, $\displaystyle\sum_{i=t+1}^{T} w(x'_i, h^*(x'_i)) \leq \frac{w(x_{t+1}, h^*(x_{t+1}))}{c(\{x_{t+1}\})} \sum_{i=t+1}^{T} c(\{x'_i\}) \leq w(x_{t+1}, h^*(x_{t+1}))$, which is due

to the fact that $\displaystyle\sum_{i=t+1}^{T} c(\{x'_i\}) = c(\{x'_i\}_{i=t+1}^T) < c(\{x_{t+1}\})$. This implies that:

$$f_{\text{worst}}(\pi_1 \text{ with budget } K) \geq f_{\text{worst}}(\pi_1 \text{ with budget } K/2).$$

Now let $x_{\pi_2}$ be the first item selected if we run $\pi_2$ with budget $K/2$. If $x_{\pi_2} \in \{x'_1, x'_2, \ldots, x'_t, x_{t+1}\}$, then $f_{\text{worst}}(\pi_1 \text{ with budget } K) \geq w(x_{\pi_2}, h^*(x_{\pi_2}))$. If $x_{\pi_2} \notin \{x'_1, x'_2, \ldots, x'_t, x_{t+1}\}$, then

$$\frac{w(x'_i, h^*(x'_i))}{c(\{x'_i\})} \geq \frac{w(x_{\pi_2}, h^*(x_{\pi_2}))}{c(\{x_{\pi_2}\})} \text{ for } i = 1, \ldots, t, \text{ and } \frac{w(x_{t+1}, h^*(x_{t+1}))}{c(\{x_{t+1}\})} \geq \frac{w(x_{\pi_2}, h^*(x_{\pi_2}))}{c(\{x_{\pi_2}\})}.$$

Thus,

$$\sum_{i=1}^{t} w(x'_i, h^*(x'_i)) + w(x_{t+1}, h^*(x_{t+1})) \geq \frac{w(x_{\pi_2}, h^*(x_{\pi_2}))}{c(\{x_{\pi_2}\})}\left(\sum_{i=1}^{t} c(\{x'_i\}) + c(\{x_{t+1}\})\right)$$

$$\geq w(x_{\pi_2}, h^*(x_{\pi_2})).$$

This is due to the fact that $\sum_{i=1}^{t} c(\{x_i'\}) + c(\{x_{t+1}\}) > K/2 \geq c(\{x_{\pi_2}\})$. Hence, $f_{\text{worst}}(\pi_1 \text{ with budget } K) \geq w(x_{\pi_2}, \overline{h}^*(x_{\pi_2}))$.

Since we always have $f_{\text{worst}}(\pi_1 \text{ with budget } K) \geq \max\{f_{\text{worst}}(\pi_1 \text{ with budget } K/2), w(x_{\pi_2}, h^*(x_{\pi_2}))\}$, from the proof of Theorem F.1, this implies:

$$f_{\text{worst}}(\pi_1 \text{ with budget } K) > \frac{1}{2}(1 - 1/e)f_{\text{worst}}(\pi_{1/2}^*),$$

and the proposition holds. $\qquad\square$

## C   Proof of Equation (5)

Let $Y_{\mathcal{D}}$ be the labels of the items in $X_{\mathcal{D}}$. We have:

$$x^* = \arg\max_{x} \delta(x \mid \mathcal{D})/\Delta c(x \mid X_{\mathcal{D}}) \qquad\qquad (\text{Definition of } x^*)$$

$$= \arg\max_{x} \frac{\min_{y \in \mathcal{Y}}\{f(X_{\mathcal{D}} \cup \{x\}, \mathcal{D} \cup \{(x,y)\}) - f(X_{\mathcal{D}}, \mathcal{D})\}}{\Delta c(x \mid X_{\mathcal{D}})} \qquad (\text{Definition of } \delta(x \mid \mathcal{D}))$$

$$= \arg\max_{x} \frac{\min_{y \in \mathcal{Y}}\{p_0[Y_{\mathcal{D}}; X_{\mathcal{D}}] - p_0[Y_{\mathcal{D}} \cup \{y\}; X_{\mathcal{D}} \cup \{x\}]\}}{\Delta c(x \mid X_{\mathcal{D}})} \qquad (\text{Definition of } f)$$

$$= \arg\max_{x} \frac{\min_{y \in \mathcal{Y}}\{1 - p_0[Y_{\mathcal{D}} \cup \{y\}; X_{\mathcal{D}} \cup \{x\}]/p_0[Y_{\mathcal{D}}; X_{\mathcal{D}}]\}}{\Delta c(x \mid X_{\mathcal{D}})}$$

$$\qquad\qquad\qquad (\text{Divide numerator by the constant } p_0[Y_{\mathcal{D}}; X_{\mathcal{D}}])$$

$$= \arg\max_{x} \frac{\min_{y \in \mathcal{Y}}\{1 - p_{\mathcal{D}}[y; x]\}}{\Delta c(x \mid X_{\mathcal{D}})}. \qquad (\text{Definition of posterior } p_{\mathcal{D}}[y; x])$$

## D   Example of Non-Submodular Cost Satisfying Triangle Inequality

If $\mathcal{X} = \{x_1, x_2, x_3\}$, we can construct a set function $c$ that is not submodular but satisfies the triangle inequality such that $c(\emptyset) = 0$, $c(\{x_1\}) = c(\{x_2\}) = c(\{x_3\}) = 1$, $c(\{x_1, x_2\}) = 2$, $c(\{x_1, x_3\}) = c(\{x_2, x_3\}) = 1.5$, and $c(\{x_1, x_2, x_3\}) = 2.5$. This function is not submodular because $c(\{x_3, x_2, x_1\}) - c(\{x_3, x_2\}) > c(\{x_3, x_1\}) - c(\{x_3\})$.

## E   Proof of Theorem 5

### E.1   Proof of Part (a)

Since $g_1$ and $g_2$ are cost-sensitively submodular w.r.t. $c$, for $A \subseteq B \subseteq \mathcal{X}$ and $x \in \mathcal{X} \setminus B$, we have:

$$\frac{g_1(A \cup \{x\}) - g_1(A)}{\Delta c(x \mid A)} \geq \frac{g_1(B \cup \{x\}) - g_1(B)}{\Delta c(x \mid B)}, \text{ and}$$

$$\frac{g_2(A \cup \{x\}) - g_2(A)}{\Delta c(x \mid A)} \geq \frac{g_2(B \cup \{x\}) - g_2(B)}{\Delta c(x \mid B)}.$$

Multiplying $\alpha$ and $\beta$ into both sides of the first and second inequality respectively, then summing the resulting inequalities, we have:

$$\frac{(\alpha g_1(A \cup \{x\}) + \beta g_2(A \cup \{x\})) - (\alpha g_1(A) + \beta g_2(A))}{\Delta c(x \mid A)}$$

$$\geq \frac{(\alpha g_1(B \cup \{x\}) + \beta g_2(B \cup \{x\})) - (\alpha g_1(B) + \beta g_2(B))}{\Delta c(x \mid B)}.$$

Thus, $\alpha g_1 + \beta g_2$ is cost-sensitively submodular w.r.t. $c$.

## E.2 Proof of Part (b)

Since $g$ is cost-sensitively submodular w.r.t. $c_1$, for all $A \subseteq B \subseteq \mathcal{X}$ and $x \in \mathcal{X} \setminus B$, we have:

$$\frac{g(A \cup \{x\}) - g(A)}{\Delta c_1(x \mid A)} \geq \frac{g(B \cup \{x\}) - g(B)}{\Delta c_1(x \mid B)},$$

which implies:

$$(g(A \cup \{x\}) - g(A))(c_1(B \cup \{x\}) - c_1(B)) \geq (g(B \cup \{x\}) - g(B))(c_1(A \cup \{x\}) - c_1(A)).$$

Multiplying $\alpha$ into both sides of this inequality, we have:

$$(g(A \cup \{x\}) - g(A))(\alpha c_1(B \cup \{x\}) - \alpha c_1(B)) \geq (g(B \cup \{x\}) - g(B))(\alpha c_1(A \cup \{x\}) - \alpha c_1(A)).$$

Similarly, we also have:

$$(g(A \cup \{x\}) - g(A))(\beta c_2(B \cup \{x\}) - \beta c_2(B)) \geq (g(B \cup \{x\}) - g(B))(\beta c_2(A \cup \{x\}) - \beta c_2(A)).$$

Summing these inequalities, we have:

$$(g(A \cup \{x\}) - g(A))(\alpha c_1(B \cup \{x\}) + \beta c_2(B \cup \{x\}) - \alpha c_1(B) - \beta c_2(B))$$
$$\geq (g(B \cup \{x\}) - g(B))(\alpha c_1(A \cup \{x\}) + \beta c_2(A \cup \{x\}) - \alpha c_1(A) - \beta c_2(A)).$$

Thus,

$$\frac{g(A \cup \{x\}) - g(A)}{(\alpha c_1(A \cup \{x\}) + \beta c_2(A \cup \{x\})) - (\alpha c_1(A) + \beta c_2(A))}$$

$$\geq \frac{g(B \cup \{x\}) - g(B)}{(\alpha c_1(B \cup \{x\}) + \beta c_2(B \cup \{x\})) - (\alpha c_1(B) + \beta c_2(B))}.$$

Hence, $g$ is cost-sensitively submodular w.r.t. $\alpha c_1 + \beta c_2$.

## E.3 Proof of Parts (c) and (d)

First, we prove the following lemma.

**Lemma E.1.** *For any integer $k \geq 1$, if $c(S) = (g(S))^k$ for all $S \subseteq \mathcal{X}$ and $g$ is monotone, then $g$ is cost-sensitively submodular w.r.t. $c$.*

*Proof.* If $k = 1$, this trivially holds. If $k \geq 2$, for all $A \subseteq B \subseteq \mathcal{X}$ and $x \in \mathcal{X} \setminus B$, we have:

$$\frac{g(A \cup \{x\}) - g(A)}{\Delta c(x \mid A)} = \frac{g(A \cup \{x\}) - g(A)}{(g(A \cup \{x\}))^k - (g(A))^k} = \frac{1}{\sum_{i=0}^{k-1}(g(A \cup \{x\}))^{k-1-i}(g(A))^i}.$$

Similarly,

$$\frac{g(B \cup \{x\}) - g(B)}{\Delta c(x \mid B)} = \frac{1}{\sum_{i=0}^{k-1}(g(B \cup \{x\}))^{k-1-i}(g(B))^i}.$$

Since $g$ is monotone, $g(A \cup \{x\}) \leq g(B \cup \{x\})$ and $g(A) \leq g(B)$. Thus,

$$\sum_{i=0}^{k-1}(g(A \cup \{x\}))^{k-1-i}(g(A))^i \leq \sum_{i=0}^{k-1}(g(B \cup \{x\}))^{k-1-i}(g(B))^i.$$

Hence, $\dfrac{g(A \cup \{x\}) - g(A)}{\Delta c(x \mid A)} \geq \dfrac{g(B \cup \{x\}) - g(B)}{\Delta c(x \mid B)}$, which implies that $g$ is cost-sensitively submodular w.r.t. $c$. $\qquad\square$

Applying part (b) and Lemma E.1, we can easily see that part (c) holds. Furthermore, from parts (b), (c), and the Taylor approximation of $e^{g(S)}$, part (d) also holds.

# F  General Theorems and Proofs for the Non-Modular Cost Setting

First, we state the theorems for the general, possibly non-modular, cost setting.

**Theorem F.1.** *Assume the utility $f$ and the cost $c$ satisfy the assumptions in Section 4. Let $\pi^*$ be the optimal policy for the worst-case adaptive optimization problem with utility $f$, cost $c$, and budget $K$. The policy $\pi$ defined by Equation* (3) *satisfies:*

$$f_{worst}(\pi) > \frac{1}{2}\left(1 - 1/e\right) f_{worst}(\pi^*).$$

**Theorem F.2.** *Assume the same setting as in Theorem F.1. Let $\pi^*_{1/2}$ be the optimal policy for the worst-case adaptive optimization problem with budget $K/2$. The policy $\pi_{1/2}$ in Section 3.2.3 satisfies:*

$$f_{worst}(\pi_{1/2}) > \frac{1}{2}\left(1 - 1/e\right) f_{worst}(\pi^*_{1/2}).$$

Now we prove the above theorems.

## F.1  Proof of Theorem F.1

Without loss of generality, we assume each item can be selected by at least one policy given the budget $K$; otherwise, we can simply remove that item from the item set. First, consider the policy $\pi_1$. Let $h_1 = \arg\min_h f(x_h^{\pi_1}, h)$ be the worst-case realization of $\pi_1$. We have $f_{\text{worst}}(\pi_1) = f(x_{h_1}^{\pi_1}, h_1)$. Note that $h_1$ corresponds to a path from the root to a leaf of the policy tree of $\pi_1$, and let the items and states along this path (starting from the root) be:

$$h_1 = \{(x_1, y_1), (x_2, y_2), \ldots, (x_{|h_1|}, y_{|h_1|})\}.$$

At any item $x_i$ along the path $h_1$, imagine that we run the optimal policy $\pi^*$ right after selecting $x_i$ and then follow the paths consistent with $\{(x_1, y_1), \ldots, (x_i, y_i)\}$ down to a leaf of the policy tree of $\pi^*$. We consider the following adversary's path $h^a = \{(x_1^a, y_1^a), (x_2^a, y_2^a), \ldots, (x_{|h^a|}^a, y_{|h^a|}^a)\}$ in the policy tree of $\pi^*$ that satisfies:

$$y_j^a = \arg\min_y \{ f(\{x_t\}_{t=1}^i \cup \{x_t^a\}_{t=1}^{j-1} \cup \{x_j^a\}, \{y_t\}_{t=1}^i \cup \{y_t^a\}_{t=1}^{j-1} \cup \{y\})$$

$$- f(\{x_t\}_{t=1}^i \cup \{x_t^a\}_{t=1}^{j-1}, \{y_t\}_{t=1}^i \cup \{y_t^a\}_{t=1}^{j-1})\}$$

if $x_j^a$ has not appeared in $\{x_1, \ldots, x_i\}$. Otherwise, $y_j^a = y_t$ if $x_j^a = x_t$ for some $t = 1, \ldots, i$. In the above, since $f$ satisfies minimal dependency, we write $f(\{x_t\}_{t=1}^i, \{y_t\}_{t=1}^i)$ to denote the utility obtained after observing $\{(x_t, y_t)\}_{t=1}^i$.

Assume we follow the path $h_1$ during the execution of $\pi_1$. Let $r$ be the number of iterations (the **repeat** loop) executed in the algorithm for $\pi_1$ (see Figure 1) until the first time an item in the corresponding adversary's path is considered, but not added to $\mathcal{D}$ due to the cost budget. Let $(x_1, y_1), \ldots, (x_l, y_l)$ be the items selected (i.e., added to $\mathcal{D}$) along the path $h_1$ until iteration $r$. Furthermore, let $x_{l+1}$ be the item in the corresponding adversary's path (i.e., the adversary's path right after selecting $x_l$) that is considered but not added to $\mathcal{D}$. Consider an arbitrary state $y_{l+1}$ for $x_{l+1}$. Also let $j_i$ be the iteration where $x_i$ ($1 \leq i \leq l+1$) is considered. For $i = 1, 2, \ldots, l+1$, define:

$$u_i = f(\{x_t\}_{t=1}^i, \{y_t\}_{t=1}^i) - f(\{x_t\}_{t=1}^{i-1}, \{y_t\}_{t=1}^{i-1}), \quad v_i = \sum_{t=1}^i u_t, \quad \text{and} \quad z_i = f_{\text{worst}}(\pi^*) - v_i.$$

We first prove the following lemma.

**Lemma F.1.** *For $i = 1, \ldots, l+1$, after each iteration $j_i$, we have $u_i \geq \dfrac{\Delta c(x_i \mid \{x_t\}_{t=1}^{i-1})}{K} z_{i-1}.$*

*Proof.* For $i = 1, \ldots, l+1$ and $j = 1, \ldots, |h^a|$ (note that $h^a$ depends on $i$), we have:

$$\frac{u_i}{\Delta c(x_i \mid \{x_t\}_{t=1}^{i-1})}$$

$$= \frac{f(\{x_t\}_{t=1}^{i}, \{y_t\}_{t=1}^{i}) - f(\{x_t\}_{t=1}^{i-1}, \{y_t\}_{t=1}^{i-1})}{c(\{x_t\}_{t=1}^{i}) - c(\{x_t\}_{t=1}^{i-1})}$$

$$\geq \min_y \frac{f(\{x_t\}_{t=1}^{i-1} \cup \{x_i\}, \{y_t\}_{t=1}^{i-1} \cup \{y\}) - f(\{x_t\}_{t=1}^{i-1}, \{y_t\}_{t=1}^{i-1})}{c(\{x_t\}_{t=1}^{i}) - c(\{x_t\}_{t=1}^{i-1})}$$

$$\geq \min_y \frac{f(\{x_t\}_{t=1}^{i-1} \cup \{x_j^a\}, \{y_t\}_{t=1}^{i-1} \cup \{y\}) - f(\{x_t\}_{t=1}^{i-1}, \{y_t\}_{t=1}^{i-1})}{c(\{x_t\}_{t=1}^{i-1} \cup \{x_j^a\}) - c(\{x_t\}_{t=1}^{i-1})}$$

$$\geq \min_y \frac{f(\{x_t\}_{t=1}^{i-1} \cup \{x_t^a\}_{t=1}^{j-1} \cup \{x_j^a\}, \{y_t\}_{t=1}^{i-1} \cup \{y_t^a\}_{t=1}^{j-1} \cup \{y\}) - f(\{x_t\}_{t=1}^{i-1} \cup \{x_t^a\}_{t=1}^{j-1}, \{y_t\}_{t=1}^{i-1} \cup \{y_t^a\}_{t=1}^{j-1})}{c(\{x_t\}_{t=1}^{i-1} \cup \{x_t^a\}_{t=1}^{j-1} \cup \{x_j^a\}) - c(\{x_t\}_{t=1}^{i-1} \cup \{x_t^a\}_{t=1}^{j-1})}$$

$$= \frac{f(\{x_t\}_{t=1}^{i-1} \cup \{x_t^a\}_{t=1}^{j}, \{y_t\}_{t=1}^{i-1} \cup \{y_t^a\}_{t=1}^{j}) - f(\{x_t\}_{t=1}^{i-1} \cup \{x_t^a\}_{t=1}^{j-1}, \{y_t\}_{t=1}^{i-1} \cup \{y_t^a\}_{t=1}^{j-1})}{c(\{x_t\}_{t=1}^{i-1} \cup \{x_t^a\}_{t=1}^{j}) - c(\{x_t\}_{t=1}^{i-1} \cup \{x_t^a\}_{t=1}^{j-1})},$$

where the first equality is from the definition of $u_i$, the second inequality is from the greedy criterion and assumption of $x_{l+1}$, the third inequality is from the pointwise cost-sensitive submodularity of $f$ and $c$, and the last equality is from the definition of $y_j^a$.

Thus,

$$z_{i-1}$$

$$= f_{\text{worst}}(\pi^*) - v_{i-1}$$

$$\leq f(\{x_t\}_{t=1}^{i-1} \cup \{x_t^a\}_{t=1}^{|h^a|}, \{y_t\}_{t=1}^{i-1} \cup \{y_t^a\}_{t=1}^{|h^a|}) - f(\{x_t\}_{t=1}^{i-1}, \{y_t\}_{t=1}^{i-1})$$

$$= \sum_{j=1}^{|h^a|} (f(\{x_t\}_{t=1}^{i-1} \cup \{x_t^a\}_{t=1}^{j}, \{y_t\}_{t=1}^{i-1} \cup \{y_t^a\}_{t=1}^{j}) - f(\{x_t\}_{t=1}^{i-1} \cup \{x_t^a\}_{t=1}^{j-1}, \{y_t\}_{t=1}^{i-1} \cup \{y_t^a\}_{t=1}^{j-1}))$$

$$\leq \sum_{j=1}^{|h^a|} \frac{u_i(c(\{x_t\}_{t=1}^{i-1} \cup \{x_t^a\}_{t=1}^{j}) - c(\{x_t\}_{t=1}^{i-1} \cup \{x_t^a\}_{t=1}^{j-1}))}{\Delta c(x_i \mid \{x_t\}_{t=1}^{i-1})}$$

$$= \frac{u_i}{\Delta c(x_i \mid \{x_t\}_{t=1}^{i-1})} \sum_{j=1}^{|h^a|} (c(\{x_t\}_{t=1}^{i-1} \cup \{x_t^a\}_{t=1}^{j}) - c(\{x_t\}_{t=1}^{i-1} \cup \{x_t^a\}_{t=1}^{j-1}))$$

$$= \frac{u_i}{\Delta c(x_i \mid \{x_t\}_{t=1}^{i-1})} (c(\{x_t\}_{t=1}^{i-1} \cup \{x_t^a\}_{t=1}^{|h^a|}) - c(\{x_t\}_{t=1}^{i-1}))$$

$$\leq \frac{u_i}{\Delta c(x_i \mid \{x_t\}_{t=1}^{i-1})} c(\{x_t^a\}_{t=1}^{|h^a|})$$

$$\leq \frac{u_i}{\Delta c(x_i \mid \{x_t\}_{t=1}^{i-1})} K.$$

In the above, the first inequality is from the definition of $v_{i-1}$ and the pointwise monotonicity of $f$, the second inequality is from the previous discussion, the third inequality is from the triangle inequality for $c$, and the last inequality is from the fact that $h^a$ is a path of $\pi^*$, whose cost is at most $K$. Thus, Lemma F.1 holds. □

Using Lemma F.1, we now prove the next lemma.

**Lemma F.2.** *For $i = 1, \ldots, l+1$, after each iteration $j_i$, we have:*

$$v_i \geq \left[ 1 - \prod_{t=1}^{i} \left( 1 - \frac{\Delta c(x_t \mid \{x_j\}_{j=1}^{t-1})}{K} \right) \right] f_{worst}(\pi^*).$$

*Proof.* We prove this lemma by induction.

**Base case:** For $i = 1$, consider the path $h^b = \{(x_1^b, y_1^b), (x_2^b, y_2^b), \ldots, (x_{|h^b|}^b, y_{|h^b|}^b)\}$ in the policy tree of $\pi^*$ that satisfies $y_i^b = \arg\min_y f(\{x_i^b\}, \{y\})$. For all $i = 1, 2, \ldots, |h^b|$, we have:

$$\frac{f(\{x_1\}, \{y_1\})}{c(\{x_1\})} = \frac{f(\{x_1\}, \{y_1\}) - f(\emptyset, \emptyset)}{c(\{x_1\}) - c(\emptyset)} \geq \frac{\min_y \{f(\{x_1\}, \{y\}) - f(\emptyset, \emptyset)\}}{c(\{x_1\}) - c(\emptyset)}$$

$$\geq \frac{\min_y \{f(\{x_i^b\}, \{y\}) - f(\emptyset, \emptyset)\}}{c(\{x_i^b\}) - c(\emptyset)} = \frac{f(\{x_i^b\}, \{y_i^b\}) - f(\emptyset, \emptyset)}{c(\{x_i^b\}) - c(\emptyset)}$$

$$\geq \frac{f(\{x_j^b\}_{j=1}^i, \{y_j^b\}_{j=1}^i) - f(\{x_j^b\}_{j=1}^{i-1}, \{y_j^b\}_{j=1}^{i-1})}{c(\{x_j^b\}_{j=1}^i) - c(\{x_j^b\}_{j=1}^{i-1})}.$$

In the above, the first equality is due to $f(\emptyset, \emptyset) = 0$ and $c(\emptyset) = 0$, the second inequality is due to the greedy criterion of $\pi_1$, the second equality is from the definition of $y_i^b$, and the last inequality is from the cost-sensitive submodularity of $f$.

Thus, for all $i = 1, 2, \ldots, |h^b|$,

$$\frac{f(\{x_1\}, \{y_1\})}{c(\{x_1\})}(c(\{x_j^b\}_{j=1}^i) - c(\{x_j^b\}_{j=1}^{i-1})) \geq f(\{x_j^b\}_{j=1}^i, \{y_j^b\}_{j=1}^i) - f(\{x_j^b\}_{j=1}^{i-1}, \{y_j^b\}_{j=1}^{i-1}).$$

Summing the above inequality for all $i$, we have:

$$\frac{f(\{x_1\}, \{y_1\})}{c(\{x_1\})} c(\{x_j^b\}_{j=1}^{|h^b|}) \geq f(\{x_j^b\}_{j=1}^{|h^b|}, \{y_j^b\}_{j=1}^{|h^b|}).$$

Since $h^b$ is a path of $\pi^*$, we have $c(\{x_j^b\}_{j=1}^{|h^b|}) \leq K$ and $f(\{x_j^b\}_{j=1}^{|h^b|}, \{y_j^b\}_{j=1}^{|h^b|}) \geq f_{\text{worst}}(\pi^*)$. Thus, $v_1 = f(\{x_1\}, \{y_1\}) \geq \frac{c(\{x_1\})}{K} f_{\text{worst}}(\pi^*)$ and the base case holds.

**Inductive step:** Now assume the lemma holds for $i - 1$. We have:

$$v_i$$
$$= v_{i-1} + u_i$$
$$\geq v_{i-1} + \frac{\Delta c(x_i \mid \{x_t\}_{t=1}^{i-1})}{K}(f_{\text{worst}}(\pi^*) - v_{i-1})$$
$$= (1 - \frac{\Delta c(x_i \mid \{x_t\}_{t=1}^{i-1})}{K})v_{i-1} + \frac{\Delta c(x_i \mid \{x_t\}_{t=1}^{i-1})}{K} f_{\text{worst}}(\pi^*)$$
$$\geq (1 - \frac{\Delta c(x_i|\{x_t\}_{t=1}^{i-1})}{K})\left[1 - \prod_{t=1}^{i-1}\left(1 - \frac{\Delta c(x_t|\{x_j\}_{j=1}^{t-1})}{K}\right)\right] f_{\text{worst}}(\pi^*) + \frac{\Delta c(x_i|\{x_t\}_{t=1}^{i-1})}{K} f_{\text{worst}}(\pi^*)$$
$$= \left[1 - \prod_{t=1}^{i}\left(1 - \frac{\Delta c(x_t \mid \{x_j\}_{j=1}^{t-1})}{K}\right)\right] f_{\text{worst}}(\pi^*),$$

where the first inequality is from Lemma F.1 and the second inequality is from the inductive hypothesis. □

Now we prove Theorem F.1. Applying Lemma F.2 to iteration $j_{l+1}$, we have:

$$v_{l+1} \geq \left[1 - \prod_{t=1}^{l+1}\left(1 - \frac{\Delta c(x_t \mid \{x_j\}_{j=1}^{t-1})}{K}\right)\right] f_{\text{worst}}(\pi^*)$$

$$\geq \left[1 - \prod_{t=1}^{l+1}\left(1 - \frac{\Delta c(x_t \mid \{x_j\}_{j=1}^{t-1})}{\sum_{i=1}^{l+1} \Delta c(x_i \mid \{x_j\}_{j=1}^{i-1})}\right)\right] f_{\text{worst}}(\pi^*)$$

$$\geq \left[1 - \left(1 - \frac{1}{l+1}\right)^{l+1}\right] f_{\text{worst}}(\pi^*)$$

$$> \left(1 - \frac{1}{e}\right) f_{\text{worst}}(\pi^*).$$

The second inequality is due to $\sum_{i=1}^{l+1} \Delta c(x_i \mid \{x_j\}_{j=1}^{i-1}) = c(\{x_1, \ldots, x_{l+1}\}) > K$, and the third inequality is due to the fact that the function $1 - \prod_{t=1}^{n}\left(1 - \frac{a_t}{\sum_{i=1}^{n} a_i}\right)$ achieves its minimum when $a_1 = \ldots = a_n = \frac{\sum_i a_i}{n}$. Hence,

$$v_l + u_{l+1} = v_{l+1} > \left(1 - \frac{1}{e}\right) f_{\text{worst}}(\pi^*).$$

Now consider the first item $x$ selected by the policy $\pi_2$. Let $y_w$ be the state of $x$ in the worst-case path of the policy tree of $\pi_2$. In the previous arguments, note that $y_{l+1}$ can be arbitrary, thus without loss of generality, we can set $y_{l+1} = \arg\min_y f(\{x_{l+1}\}, \{y\})$. Now we have:

$$\begin{aligned}
f_{\text{worst}}(\pi_2) &\geq f(\{x\}, \{y_w\}) \geq \min_y f(\{x\}, \{y\}) \\
&\geq \min_y f(\{x_{l+1}\}, \{y\}) = f(\{x_{l+1}\}, \{y_{l+1}\}) \\
&\geq \frac{f(\{x_t\}_{t=1}^{l+1}, \{y_t\}_{t=1}^{l+1}) - f(\{x_t\}_{t=1}^{l}, \{y_t\}_{t=1}^{l})}{c(\{x_t\}_{t=1}^{l+1}) - c(\{x_t\}_{t=1}^{l})} c(\{x_{l+1}\}) \\
&\geq u_{l+1},
\end{aligned}$$

where the first inequality is from the pointwise monotonicity of $f$, the third inequality is from the greedy criterion of $\pi_2$, the fourth inequality is from the pointwise cost-sensitive submodularity of $f$, and the last inequality is from the triangle inequality for $c$.

Furthermore, $f_{\text{worst}}(\pi_1) \geq v_l$ due to the pointwise monotonicity of $f$ and $v_l$ is computed along the worst-case path of $\pi_1$. Hence,

$$f_{\text{worst}}(\pi_1) + f_{\text{worst}}(\pi_2) > \left(1 - \frac{1}{e}\right) f_{\text{worst}}(\pi^*).$$

Therefore, $f_{\text{worst}}(\pi) = \max\{f_{\text{worst}}(\pi_1), f_{\text{worst}}(\pi_2)\} > \frac{1}{2}(1 - \frac{1}{e}) f_{\text{worst}}(\pi^*)$.

From this proof, we can easily see that the theorem still holds if we replace the policy $\pi_2$ with only the first item $x$ that it selects. In other words, we can terminate the policy $\pi_2$ right after it selects the first item and the near-optimality is still guaranteed.

### F.2 Proof of Theorem F.2

Let $h_{1/2} = \arg\min_h f(x_h^{\pi_{1/2}}, h)$ be the worst-case realization of $\pi_{1/2}$. We have $f_{\text{worst}}(\pi_{1/2}) = f(x_{h_{1/2}}^{\pi_{1/2}}, h_{1/2})$. Note that $x_{h_{1/2}}^{\pi_{1/2}} = x_{h_{1/2}}^{\pi_1} \cup x_{h_{1/2}}^{\pi_2}$, where $x_{h_{1/2}}^{\pi_1}$ and $x_{h_{1/2}}^{\pi_2}$ are the sets selected by $\pi_1$ and $\pi_2$ respectively in the policy $\pi_{1/2}$ under the realization $h_{1/2}$. Thus, $f_{\text{worst}}(\pi_{1/2}) \geq f(x_{h_{1/2}}^{\pi_1}, h_{1/2})$ and $f_{\text{worst}}(\pi_{1/2}) \geq f(x_{h_{1/2}}^{\pi_2}, h_{1/2})$ due to the pointwise monotonicity of $f$.

From the definition of $f_{\text{worst}}$, we have $f(x_{h_{1/2}}^{\pi_1}, h_{1/2}) \geq \min_h f(x_h^{\pi_1}, h) = f_{\text{worst}}(\pi_1)$. Hence, $f_{\text{worst}}(\pi_{1/2}) \geq f_{\text{worst}}(\pi_1)$. Similarly, $f_{\text{worst}}(\pi_{1/2}) \geq f_{\text{worst}}(\pi_2)$. From Theorem F.1, either $f_{\text{worst}}(\pi_1)$ or $f_{\text{worst}}(\pi_2)$ must be greater than $\frac{1}{2}(1 - 1/e) f_{\text{worst}}(\pi_{1/2}^*)$. Therefore, Theorem F.2 holds.