[Reviews · NeurIPS 2016]

Reviewer 1

Summary

This paper considers the worst-case adaptive optimization problem with budget constraint. The utility function is assumed to be monotone submodular and the constraint can be modular or satisfy the triangle inequality. The authors show that under the assumptions that the utility function is 1) pointwise submodular, 2) pointwise monotone and 3) the value of the function depends only on the items selected so far, and the cost function is modular then the combination of two simple greedy algorithms provided a bicriteria approximation (Theorem 4). They also have (in my opinion) generalized their results to the case where the cost satisfies a triangle inequality. Despite author's intention, I do not find Theorem 5 significant.

Qualitative Assessment

There are many references missing. The authors state "In contrast to previous works on adaptive optimization with budget constraint (both in the average and worst cases), we consider not only modular cost functions but also general, possibly non-modular,cost functions on sets of decisions." They should cite the relevant papers. Online Submodular Set Cover, Ranking, and Repeated Active Learning Active Learning and Submodular Functions (Guillory's Phd thesis which should definitely be cited) Near Optimal Bayesian Active Learning for Decision Making They consider only MONOTONE submodular functions. But there has been some recent work on the non-monotone case. I think it would be more interesting (and very plausible) to see whether their method generalizes to this setting as well. The corresponding reference that I am aware of (and I think the authors should cite) Nonmonotone adaptive submodular maximization

Confidence in this Review

3-Expert (read the paper in detail, know the area, quite certain of my opinion)


Reviewer 2

Summary

The authors analyzed the problem of worst-case adaptive optimization with budget constraint. Crucially, the allow the cost to be either modular or non-modular and the utility function to satisfy either the pointwise submodularity condition or the pointwise cost-sensitive submodularity condition. They then present a combined near-optimal policy in the more general case.

Qualitative Assessment

The work is of interest from a theoretical and practical point of view. In the experimental part the authors analyze the performance of 1) the cost-insensitive greedy policy (LC) and of 2) cost-average greedy policy (ALC) and of 3) their proposed budgeted least confidence (BLC). The paper would benefit from a stronger baseline to compare against and from reporting a significance notion (i.e. the standard deviation). As for the baseline, although not guaranteed to work near optimally in theory, one could just report the empirical performance of running the two algorithms for a fraction k of the budget and then choosing the best for the remaining part of the budget for various values of k.

Confidence in this Review

1-Less confident (might not have understood significant parts)


Reviewer 3

Summary

The paper generalizes adaptive submodular optimization to some cases where the cost is not modular.

Qualitative Assessment

Potential impact or usefulness: Generalizing adaptive submodular optimization (or even non-adaptive) for non-modular costs is an interesting problem. However, the requirement of cost-sensitive submodularity is pretty strict, which makes the results less impressive. The proofs for this limited setting are still novel and the techniques might lead to results for more general cost/utility functions. Technical quality: The proof techniques are interesting. The experiments show that BLC sometimes protects against the poor results obtained by one of LC and ALC (but not always -- Cost = R1 is pretty bad for BLC). The costs are very synthetic though, so it's not clear to me how much better BLC would be compared to LC and ALC under different costs. Novelty/originality: The concept of cost-sensitive submodularity is interesting, but it's just so strict that it seems like very few useful problems will have cost-sensitive submodularity without just having a modular cost. I think that this paper would be very strong if either of these can be shown: (1) the combined policy also works in some settings more general than cost-sensitive submodularity (2) for problems without cost-sensitive submodularity, there cannot exist reasonable (polynomial-time?) algorithms with approximation guarantees Clarity and presentation: Overall, the writing was easy to understand. I only have two small complaints. First, section 3.3 felt out of place. I think it would be more natural if it were right after section 2 (or even merged into section 2) or with the experiments. Second, the paper seemed to imply that the methods would be able to handle any cost function (whether modular or not), but in section 4, it became clear that there were still restrictions on the cost function.

Confidence in this Review

2-Confident (read it all; understood it all reasonably well)


Reviewer 4

Summary

The paper addresses the problems of adaptive optimization with budget constraint. While previous studies address this problem for the average case, this study focuses on the worst case. In contrast to previous works on adaptive optimization with budget constraint, the authors of this paper consider not only modular cost functions but also general, possibly non-modular, cost functions. The authors investigate the near-optimality of two greedy policies for the worst-caseadaptive optimization problem with budget constraint (a policy is near-optimal if its worst-case utility is within a constant factor of the optimal worst-case utility). The authors prove that in the general case, these two policies cannot achieve near-optimality with non-uniform modular costs, but the best between these two greedy policies always achieves near-optimality. Following that, the main result of the paper is derived: The two greedy policies into one greedy policy that is near-optimal with respect to the optimal worst-case policy that uses half of the budget. In addition, the authors propose a novel class of utility functions satisfying a property that is a generalization of cost-sensitive submodularity to the adaptive setting. For this new class of utilities, the authors prove similar near-optimality results for the greedy policies as in the case of modular costs. The proposed algorithm is compared experimentally to other algorithms, and results show that even though it does not yield the best result in almost any case (which is not surprising and supported by the theoretical results in the paper), it is a robust choice for an algorithm that achieves the second best results in all cases.

Qualitative Assessment

Definitions and mathematical proofs are concise and easy to understand. Related work is covered in a sufficient manner. The paper exhibits a novel mathematical analysis relevant to adaptive optimization with budget constraint, a problem with many practical implications. The setting in which the analysis is given is wider compared to previous papers on the subject. Two greedy algorithms are analyzed, and despite negative results regarding their near-optimality, results (proofs) show that a combination of the two has interesting properties and can be sound and useful in some cases. In addition, a subset of utility functions under which strong positive results can be proved is correctly identified. The proposed algorithm is derived naturally from the theoretical results, and experiment results support the theoretical results. On the down side, the proposed algorithm does not perform better than other algorithms (but also never performs catastrophically bad), and was not examined in many different experiment settings.

Confidence in this Review

1-Less confident (might not have understood significant parts)


Reviewer 5

Summary

This paper discussed the pointwise submodular maximization problem with non-uniform costs. In previous work, only the case of uniform costs is considered. This paper treats both cases of linear and nonlinear cost function. For the nonlinear case, the authors propose a new condition, cost-sensitive pointwise submodularity. The proposed algorithm that can be applied to both cases achieves (1-1/e)/2-approximation in comparison to the optimal policy with a half budget. The authors conducted experiments about pool-based active learning on the benchmark datasets, and show their proposed method is better than either the greedy policy or the cost-sensitive greedy policy.

Qualitative Assessment

This paper would offer good contributions to research on pointwise submodularity, but it lacks extensive analyses in some parts such as the following: - The authors should refer to existing work on the non-adaptive submodular maximization subject to a knapsack constraint, especially the following Sviridenko's paper. Maxim Sviridenko. A note on maximizing a submodular set function subject to a knapsack constraint. Operations Research Letters 32(1): 41-43, 2004. In this paper, it was showed that we can achieve (1-1/e)-approximation with the partial enumeration technique. The authors should discuss whether it is possible to extend this approach to the adaptive setting. - Line 337: The sentence contains an error. In Khuller-Moss-Naor's paper, they proposed both (1-1/\sqrt{e})-approximation and (1-1/e)-approximation algorithms, but not (1-1/e)/2-approximation algorithm. Is it possible to extend their results to the adaptive setting? The authors should investigate this approach. - This paper studies on the cases of ``the cost function suffices the triangle inequality'', i.e., it holds c(X) + c(Y) \ge c(X \cup Y) for all X and Y. This condition is called subadditiveness, and there are several work studying the submodular maximization with a subadditive cost function such as Soma-Yoshida, NIPS2015. The authors should compare their results with these existing work on subadditive costs. - In the experiments, the authors claim that the proposed method is stable, i.e., better than the worse one among the greedy policy and the cost-sensitive greedy policy. It is true but for some datasets the proposed method is much worse than the better one of them, and seems not so practical. From the above reasons, this paper does not achieve the standard quality of NIPS in both theoretical and practical aspects.

Confidence in this Review

2-Confident (read it all; understood it all reasonably well)


Reviewer 6

Summary

The paper studies adaptive optimization when in the case of modular and sub-modular cost and utility functions. Theoretical results on non-optimality of 2 greedy algorithms are presented. Then, it is shown that a combination of these 2 is near optimal under certain assumptions. Some experiments are provided.

Qualitative Assessment

This is a promising paper that deals with a relevant problem. In the introduction, although the sensor running example is provided, it would be nice to illustrate cost/utility in context/with examples earlier and with more clarity. For instance, these "hooks" are provided later in the paper, in the applications section -- I would put some of them in the introduction. The combined policy, is not provided in pseusocode. It would be nice to have it in pseudocode, since it is the algorithm that the paper proposes. There is plenty of room for improvement in the experiments. For instance, an experiment illustrating benefits in the "sensor" example would make the paper stronger. It also seems that an experiment in a real active learning experiment for text classification (say, using with real human annotators) could be relevant. Finally, I would highlight/give more details on the benefits/implications of using BLC further in the introduction and conclusion.

Confidence in this Review

2-Confident (read it all; understood it all reasonably well)